# Optimized deep learning approach for lung cancer detection using flying fox optimization and bidirectional generative adversarial networks



Manal Abdullah Alohali[1], Hamed Alqahtani[2], Shouki A. Ebad[3], Faiz Abdullah Alotaibi[4], Venkatachalam K.[5] and Jaehyuk Cho[6]

[1] Department of Information Systems, Princess Nourah Bint Abdulrahman University, Riyadh, Saudi Arabia
[2] Department of Information Systems, King Khalid University, Abha, Saudi Arabia
[3] Center for Scientific Research and Entrepreneurship, Northern Border University, Arar, Saudi Arabia
[4] Department of Information Science, King Saud University, Riyadh, Saudi Arabia
[5] Department of Software Engineering, Jeonbuk National University, Jeonju-si, Republic of Korea
[6] Department of Software Engineering and Division of Electronics & Information Engineering, Jeonbuk National University, Jeonju-si, Republic of Korea

Corresponding author
Jaehyuk Cho, chojh@jbnu.ac.kr

## ABSTRACT

Lung cancer remains one of the most prevalent and life-threatening diseases, often diagnosed at an advanced stage due to the challenges in early detection. Contributory factors include genetic mutations, smoking, alcohol consumption, and exposure to hazardous environmental conditions. Computer-aided diagnosis (CAD) systems have significantly improved early cancer detection, but limitations such as high-dimensional feature sets and overfitting issues persist. This study presents an optimised deep learning approach for lung cancer classification, integrating flying fox optimization (FFXO) for feature selection and bidirectional generative adversarial networks (Bi-GAN) for classification. The methodology consists of three key phases: (1) Data preprocessing, where missing values are handled using the multiple imputations by chain equation (MICE) technique and feature scaling is applied using standard and min-max scalers; (2) Feature selection, where the FFXO algorithm reduces feature dimensionality to enhance classification efficiency; and (3) Lung tumor classification, utilizing Bi-GAN to improve predictive accuracy. The proposed system was evaluated using key performance metrics—accuracy, precision, recall, and F1-score—and demonstrated superior performance to conventional models. Experimental results on a publicly available lung cancer dataset showed an accuracy of 98.7% highlighting the approach's robustness in precise lung tumor classification. This study provides a novel framework for improving the reliability and efficiency of lung cancer detection, offering significant potential for clinical applications.

## INTRODUCTION

Lung cancer, also referred to as pulmonary cancer, ranks as one of the most dangerous and destructive malignancies, with an annual mortality rate surpassing 75% approximately of all cancer-related deaths (*Dodia, Annappa & Mahesh, 2022*). Since timely detection is essential in determining the sustenance of an individual, pulmonary cancer has been reported to be a life-threatening and incurable illness. Lung cancer is discovered only in a more severe form in the majority of cases (*Riquelme & Akhloufi, 2020*). On the other hand, there may be an increased likelihood of recovery if discovered in earlier phases. The leading cause for challenging the early detection of pulmonary cancer is the presence of minuscule patches similar to lumps developing within the lung (*Tan, Bates & Matthew Kinsey, 2022*).

People's lifestyles have changed as a result of ongoing technological advancements for a variety of statistical reasons. The genomic and cellular layout and operation fluctuate in response to adjustments in human lifestyle (*Li et al., 2023*; *Vivek et al., 2018*). The mutation process is influenced by several circumstances, including the consumption of alcohol, cigarette usage, arsenic fiber inhalation, and radon emission contamination, leading to the incorrect mutation of a DNA cell, which ultimately results in the production of tumors (*Vivek et al., 2022*; *Zareian & Rezaei, 2022*). According to the North American Association of Central Cancer Registries (NAACCR) study, based on a recent survey, 256,178 people were affected by lung cancer in 2024. Additionally, out of 256,178 analyses conducted globally, 168,257 were found to be potentially fatal (*Lu et al., 2020*). It is concluded that among the most severe illnesses, lung cancer exhibits a variety of symptoms, including breathing difficulties, spitting up blood, vocal modifications, tightness in the chest, and changes in mucous pigment (*Cai et al., 2023*).

Diagnosing this medical condition is challenging, and several diagnostic procedures conducted on the patients are typically needed to get a precise diagnosis. This may prompt us to utilize analytical tools intended to support physicians in making judgments (*Huang et al., 2023*). The lung's neoplastic development occurs when cells within the pulmonary layers expand out of control. Through tumor progression, this proliferation may propagate outside of the lung and into surrounding cells or different regions of the body (*Yu et al., 2020*). The perpetual consumption of tobacco accounts for a significant number of lung cancer cases. About a minimal number of instances of lung cancer are found in individuals who have never used tobacco products (*Vivek et al., 2023*; *Khan & Lee, 2021*). Hereditary predispositions combined with prolonged contact with smoke from other people, asbestos-containing materials, and radioactive substances are frequently the primary cause of these kinds of illnesses (*Said et al., 2023*).

A microscopic examination is used to determine the accuracy of the diagnosis, typically done under computed tomography or airway bronchial supervision. An unprecedented illness, lung cancer causes 1.78 million deaths worldwide each year (*Han et al., 2019*; *Maheswaran et al., 2024*). An early analysis of the malignancy usually results in a higher survival rate. Because of this, it is critical to disclose pulmonary carcinomas as soon as possible. This allows for the practical analysis of eighty percent of individuals at an earlier stage of cancer development (*Rocha, Cunha & Mendonça, 2020*).

Lung cancer can also be identified by the following symptoms: extreme fatigue, tenderness in the joints, diminished appetite, broken bones, impairment of memory, loss of muscle mass, migraines, hemorrhage, brain damage, coagulation of blood, and fluid retention in the face (*Yuan, Wu & Dai, 2023*). Medical physicians have examined these symptoms using a spirometer, which gauges the amount of oxygen present in the breath of the patients. This information aids in determining the presence of lung tumors by applying several diagnostic techniques, including reflexive evaluation, fluid specimen collection, and mutation testing. A sample of blood is also tested to forecast infections within the chest. Recommendations and indicators for an early diagnosis of lung cancer are provided by the National Institute for Health and Care Excellence (*Primakov et al., 2022*). Although screening techniques have improved the identification and diagnosis of lung cancer, it is challenging to sustain faster and more reliable identification of the lung disease.

Experimental techniques are used in algorithmic learning-based computational intelligence methods to identify trends in massive databases and incrementally improve the performance of the models with further data. These algorithms are typically utilized to access various forms of information obtained consistently from multiple platforms in different areas and applications, such as business premises, defense, finance, web-based existence, and deception identification (*UrRehman et al., 2024*). Deep learning techniques have been helpful in disease diagnosis applications in recent years.

Deep learning algorithms have become highly effective tools for diagnosing lung cancer, among other healthcare analysis tasks. Convolutional neural networks (CNNs), in particular, are deep learning models skilled at identifying intricate structures and characteristics in medical data (*Khademi et al., 2023*). This capacity is essential for detecting minute irregularities that point to lung malignancy. By automating the screening procedure, deep learning models can lower the possibility of manual oversight and guarantee reliability in decision-making. Large volumes of information can be used to train deep learning models, which helps them to identify correlations and gradually get better at what they do.

Feature selection is essential to optimise deep learning models for early lung cancer diagnosis. It entails selecting the most pertinent attributes from a sizable dataset, which can significantly increase comprehension, lower computing costs, and improve the model's effectiveness. By revealing the most crucial characteristics of lung cancer diagnosis, feature selection can shed light on the fundamental physiological processes at play. Identifying the most predictive traits can help develop increasingly focused and economically feasible therapies (*Xiao et al., 2020*). Researchers can increase the efficiency of models, lower computing expenses, improve comprehensibility, and boost the overall therapeutic value of produced models by carefully choosing the most meaningful characteristics.

Bio-inspired algorithms, which draw inspiration from biological operations and events that occur naturally, are becoming increasingly popular in several fields, including feature selection. Particle swarm optimization and other bio-inspired techniques, like genetic algorithms, are skilled at navigating an ample space for exploration without being trapped in local optimal conditions (*Zhang et al., 2021b*). This is especially important for feature selection when the objective is to identify the best subset of characteristics from what could

be a significant population. Intricate, chaotic interactions between factors and the goal parameter are present in many real-world issues associated with feature selection. Bio-inspired algorithms are well-suited for various applications since they can easily manage these complications. Bio-inspired algorithms can be tailored to meet different feature selection requirements, like harmonizing feature significance and replication or minimizing both (*Kipkogei et al., 2021*).

This work uses a deep learning-based method to examine and diagnose lung cancer data. The proposed research adopts bidirectional generative adversarial networks (Bi-GAN) to achieve this. The Flying Fox Optimization (FFXO) algorithm is employed to optimize the Bi-GAN prediction model to improve its performance. The FFXO algorithm combined with the Bi-GAN model produces better predictions and diagnoses with greater precision. Ultimately, the effectiveness of the suggested algorithm is evaluated by applying it to a database of lung cancer samples.

## Research contributions

The main contributions of this research are:

1. To employ effective pre-processing strategies to handle missing values using multiple imputations by chain equations and scale the features using standard and min-max scaler techniques.
2. To incorporate a metaheuristic bio-inspired optimization algorithm such as FFXO to choose the most optimal features that best enhance the classifier model's reliability.
3. To implement Bi-directional Generative Adversarial Networks to perform lung tumor classification based on the features selected by the FFXO algorithm and demonstrate its performance efficiency by comparing it against the traditional classification algorithms.

This study is motivated by the necessity to solve these problems with an original feature selection and classification deep learning approach. Even though the diagnosis of lung cancer with deep learning models has greatly improved, their efficiency is often limited by high dimensional data and poor feature selection models. At the same time, other models usually have to deal with the datasets' limitations, making them overfitted. Many researchers have tried to enhance the accuracy of classification with the help of feature selection and optimization methods. Unfortunately, most traditional methods, including PCA and recursive feature elimination (RFE), do not consider non-linear relationships among features. To address these shortcomings, the research utilises a novel hybrid approach that combines Bi-GAN based classification and Flying Fox Optimization (FFXO) feature selection.

However, there are still significant gaps that need to be addressed:

- In adequate feature selection problem in deep learning algorithms—Many deep learning algorithms completely ignore feature selection and accept high-dimensional input data, which leads to overfitting and a lack of interpretability.

- Difficulties with small and highly imbalanced datasets—While deep learning models are proven to perform well with high volumes of data, their biases are evident with the limited medical datasets, making them generalize poorly with high variance.
- Impractical lung cancer classification in real-time—The computational inefficiency of existing hybrid models is that high-performing models, such as LungX-Net and adaptive optimization techniques, require a substantial amount of resources which are not feasible for real-life scenarios.
- Deep learning frameworks' lack of generalization—Existing models have subpar performance on different datasets because of the limited diversity of the dataset and inefficient feature selection processes.

This study aims to effectively fill the identified gaps by offering an optimised deep learning framework that incorporates:

- Adversarial training that improves feature representation and generalisation on small datasets is applied to lung cancer classification processes through Bi-GAN.
- FFXO enables better feature selection by applying the reduction of dimensionality technique while retaining important attributes, increasing the model's interpretability and minimising overfitting.
- Class imbalance is countered through synthetic data generation using Bi-GAN, improving generalisation performance and boosting model robustness in these critical areas.
- Combining the elements above will produce a hybrid model with low computational expense that is accurate and deployable in clinical settings.

### Paper organization

The remainder of the article is organised as follows. "Related Works" discusses the existing techniques proposed in the literature for classifying lung tumors using AI-based methods. "Proposed Methodology" provides a detailed description of the proposed methodology, which involves data pre-processing, selecting significant features, and categorising lung cancer. "Results and Discussion" presents the results obtained from experiments performed using the lung cancer dataset on the proposed methodology and draws a comparison with other state-of-the-art techniques. "Justification for Model Type Used" concludes the present research.

## RELATED WORKS

This section elaborates on the current works on identifying and diagnosing lung cancer using computational intelligence methods. A methodology for lung tumor identification that uses a GLCM-based exterior characterization to depict the diversity of lumps and a deep neural network to train the characteristics of edges is suggested by *Barbouchi et al. (2023)*. This approach models the structure and form based on the data learned by the deep model at the preliminary level. *Weikert et al. (2023)* investigated a computer-aided

diagnosis approach that utilized a convolutional neural network for lung cancer classification. While the transfer learning approach employs small sets of medical images, deep learning often requires enormous data sets to create solutions. With the aid of the transfer learning technique, the proposed methods were effectively developed. The highest accuracy results achieved by this technique were 97.58%.

To overcome the drawbacks of the traditional linear discriminant analysis approach, the authors in *Shin et al. (2020)* suggested the Euclidean-based linear discriminant analysis approach. Phase segmentation is carried out by connecting a support vector machine that carries out multi-class classification. The evaluation of the proposed model showed that compared to all other conventional processes, this algorithm achieves better outcomes comparably with exceptional precision and robustness.

The effectiveness and precision of the diagnostic methods are increased due to a critical evaluation of artificial neural networks for lung nodule categorization conducted in *Wang et al. (2022)*. However, the level of computation required is not reduced. Based on the histopathological inspection, tumor cells are considered one of the most significant indicators for early diagnosis in individuals with malignancy. On the other hand, computerized methods for image manipulation may reduce accidental mistakes and increase the precision of illness detection. In *Mukherjee et al. (2020)*, the researchers presented various computational methodologies based on neural networks with convolutional structures. An independent workflow was successfully built to categorize histopathological examinations across various tumor varieties. However, classifying the different forms of cancer falls short of minimizing computational expenditure.

The investigators in *Katase et al. (2022)* have suggested a different approach to identify lung tumors before symptoms appear by utilizing radial basis function neural networks with projective transformations, which in turn produced minimal average square errors as well as an excellent rate of classification. However, there had been less enhancement in data extraction efficiency. In another work (*Shetty, Jayadevappa & Tunga, 2022*), the choice of features and simultaneous classification protocols to improve disease classification accuracy are examined; however, they did not reduce classification latency.

The researchers in *Mkindu, Wu & Zhao (2023)* created a computer-aided design technique using significant segments from a series of cardiac CT sections to identify lung cancer. The pulmonary cells were separated from the cardiac CT scans using the accelerated stochastic snake approach, and the dataset was pre-processed using a filtering approach. Utilising the histogram-based segmentation technique, the region of interest was retrieved, and distinct characteristics related to form and appearance were extracted from each region identified. A fully connected deep neural network was used to train the characteristic arrays. The system was evaluated using 1,589 CT scans, and the results showed an accuracy of 95.85%.

The authors presented a Modified Grey Wolf Optimisation algorithm for feature selection in *Sun, Pang & Li (2023)* to diagnose the early stages of lung cancer disease signs. The traditional GWO technique, which draws its inspiration from the foraging habits of wolves, has been improved to choose the best characteristic subgroup more quickly while

maintaining divergence. This feature selection algorithm was utilised to evaluate the significance of every characteristic using emulated feature values. Classifiers, such as logistic regression, Bayesian classifier, and decision tree, were employed to assess the selected feature subset. With a maximum accuracy of 96.89%, the combination of modified GWO with logistic regression predicted lung tumor illness, enabling the patients to receive efficient therapies early on.

Using X-ray examination, the investigators in Chen et al. (2021) developed a classification system that uses bio-inspired methodologies to identify lung illnesses, specifically influenza, lung sclerosis, and carcinoma. Possible degraded tissues are chosen using the Artificial Ant Colony algorithm based on pixel properties: color, intensity, and luminance. Deep learning methods such as multi-layer perceptrons were used to categorise the lung tumors.

Table 1 provides a comparative analysis of various AI-based techniques for lung cancer detection, highlighting their advantages and limitations. Each study is assessed based on the techniques employed, inferences drawn, and observed limitations. The comparison shows that while deep learning models improve classification precision, some methods suffer from high computational costs, lack of effective feature selection, or increased false positive rates. This analysis underscores the need for an optimized approach like Bi-GAN with FFXO, which aims to overcome these challenges.

The notable increase in the use of hybrid deep learning models stems from their capability to use different data sources, enhance features, and boost classification performance. One such powerful model is the maximum sensitivity neural network (MSNN) which utilises transfer learning to feature extraction in lung cancer detection better. According to Lakshmi (2024), MSNN had an accuracy of 98% and 97% sensitivity, which are considerable improvements in the early cancer detection rates. This model can identify critical features from medical images and is ideal for clinical use. However, the major disadvantage of MSNN is its dependence on a vast amount of labeled data often absent in real-world medical scenarios.

Another sophisticated hybrid model, LungX-Net, combines computed tomography (CT) scans and histopathological images to achieve better classification results. Using a convolutional block attention module (CBAM) for feature extraction (Raja Rao et al., 2024), achieved recognition accuracies of 99.09% and 99.30% for CT and histopathological images, respectively. The CBAM mechanism allows the model to concentrate on the most informative parts of the lung images, thereby enhancing the differentiation of benign from malignant tumors. LungX-Net has high accuracy but has severe computational limitations, such as needing top-tier GPUs for effective processing and extensive preprocessing for multimodal data fusion. Ensemble learning techniques have been added to improve the accuracy of detecting lung cancer, which integrate multiple classifiers to enhance accuracy and reduce misclassification.

It can result in severe damage to the brain, which is frequently due to high blood pressure, brain aneurysms, or vascular malformations (Zhang et al., 2021a; Yao et al., 2024). The predictive model employs a nucleotide language model designed to identify

**Table 1 Comparison of existing works on lung cancer diagnosis.**

| References | Techniques used | Inferences | Limitations |
|---|---|---|---|
| *Huang et al. (2023)* | Deep convolutional neural networks | Improves precision of classification while utilizing less computing time. | There was no appropriate feature selection conducted. |
| *Yu et al. (2020)* | Artificial neural network | Enhanced lung nodule identification and diagnosis. | The classification accuracy rate is uncertain. |
| *Khan & Lee (2021)* | Radial basis function networks | Generates more precise results when detecting lung tumors. | There was no reduction in the false positive rate. |
| *Han et al. (2019)* | Multi-view convolutional neural networks | Boost classification precision with minimal mean square error. | The feature extraction procedure failed to perform more effectively. |
| *Rocha, Cunha & Mendonça (2020)* | Ensemble learning | Improves the classification accuracy of imbalanced medical information. | The time taken to perform classification is high. |
| *Primakov et al. (2022)* | Weight optimized neural network | Lessens processing time and improves classification reliability. | Computational complexity seems to be higher. |
| *UrRehman et al. (2024)* | Fully convolutional networks | Precise illness diagnosis was obtained using a reduced feature set. | The diagnosis process involving multiple layers was not completed in the shortest time. |
| *Khademi et al. (2023)* | Proportion-support vector machine | Detailed diagnosis, course of therapy, and results were presented. | Computing expenses were high. |
| *Xiao et al. (2020)* | Recurrent neural networks | An increased true positive rate was achieved. | The error rate was not adequately decreased. |
| *Kipkogei et al. (2021)* | Autoencoders | Improved diagnosis precision while reducing false positive rate and time taken for classification. | The issue of classification with multiple labels was not resolved. |

sequence motifs associated with m7G modifications (*Li et al., 2024a*; *Su et al., 2022*). Specific molecular markers of breast cancer can be detected, which increases the accuracy of the diagnosis (*Pu et al., 2024*). In the area of disease, phenotype classification may entail differentiating patients depending on the stage or type of disease they are suffering from? (*Li et al., 2024b*). This can allow for better quality images to be made which helps in determining the progression of the disease and the effectiveness of treatment of the retina (*Jia, Chen & Chi, 2024*; *Li et al., 2024c*). The clearer the ECG signal is, the greater the reliability of the analysis for the detection of ae abnormalities or diseases of the heart and blood vessels (*Bing et al., 2024*; *Wang et al., 2024*). The aggregated information might require cleaning or normalization before system processing (*Song et al., 2024*; *Zhu, 2024*). ENC1 has been shown to be involved in several processes in the cell, such as migration, invasion, and proliferation, which are all relevant to the progression of cancer (*Cao et al., 2024*; *Fu et al., 2024*).

One example is Adaptive Optimization algorithm which modifies model parameters in real-time to improve diagnostic accuracy (*Saxena et al., 2025*). The proposed an ensemble-based adaptive system that combines several classifiers, achieving an accuracy of 99 percent. This technique, real-time hyperparameter adjustment, balances classification accuracy and the associated resource expenditure. The most helpful aspect of this approach is that it has the potential to handle various datasets, which is very useful. Unfortunately,

the associated computational overhead with evaluating multiple models can be a constraining factor, especially within resource-limited medical facilities.

Besides, the SVRF model, a hybrid classifier that integrates support vector machine (SVM) with random forest (RF), has also proved effective. *Asif, Wang & Xu (2024)* recorded average accuracy of 91.19%. It confirms the effectiveness of the combination of statistical learning and tree-based learning for lung cancer detection. SVM is useful for mapping high dimensional features while RF learns an accurate decision boundary, which makes the combination very useful. On the contrary, this model relies heavily on feature engineering and parameter tuning, which is not ideal for cross-diverse dataset scalability.

## Motivations of current research

Conventional approaches, such as radiographic procedures, cannot always identify minor irregularities, and human comprehension is sometimes arbitrary and prone to inaccuracies. Deep learning models present an intriguing solution because of their capacity to identify intricate relationships, and automate the diagnostic procedure. Enhanced Flying Fox Optimisation (FFXO) is intended to improve model performance by resolving issues such as overfitting and premature convergence. By producing synthetic data, Bi-GANs can assist in addressing data restrictions and class disparities. By tackling these issues, the present research aims to enhance the reliability of lung cancer detection, improving clinical results and reducing the rate of death.

## PROPOSED METHODOLOGY

This section presents the workflow of the proposed system for detecting lung cancer using bio-inspired feature selection and deep learning-based classification algorithms. The dataset was preprocessed using multiple imputations by chain equation (MICE) for missing value imputation, followed by Standardization and Min-Max Normalization for feature scaling. Feature selection was performed using FFXO to reduce dimensionality while retaining the most relevant attributes. The dataset was split into 80% training and 20% testing, with 5-fold cross-validation applied to ensure robustness. The Bi-GAN model was trained using adversarial learning, where the generator created synthetic lung cancer samples while the discriminator improved classification performance. The model was optimised using Adam optimiser with a learning rate 0.0001 and trained for 100 epochs with batch normalization. Performance was evaluated using accuracy, precision, recall, and F1-score, and compared against state-of-the-art deep learning models and metaheuristic optimization techniques to validate its effectiveness. To ensure computational feasibility, the experimental setup was implemented using Python, TensorFlow, and Scikit-learn on a system with Intel Core i5, 8 GB RAM, and NVIDIA GTX 1650 GPU.

The proposed workflow, Fig. 1 illustrates the systematic approach adopted in this study, comprising three primary phases: preprocessing, feature selection, and classification. In the preprocessing phase, missing values are handled using the MICE technique, and feature scaling is applied using standardization and min-max normalization to ensure consistency in data representation. The feature selection phase employs the FFXO algorithm, which selects the most significant features, reducing dimensionality and improving classification

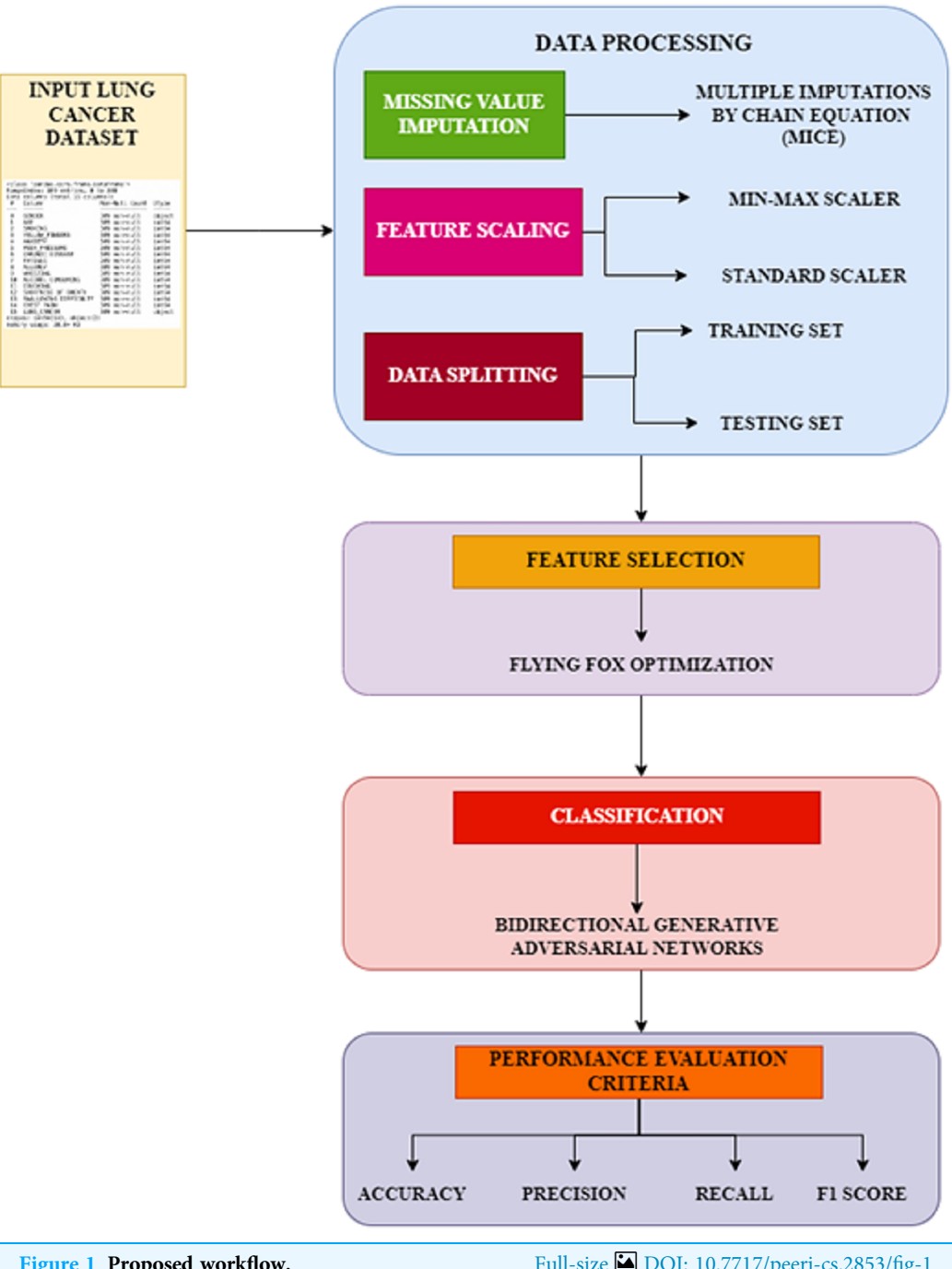

**Figure 1 Proposed workflow.**

performance. Finally, in the classification phase, a Bi-GAN is used for lung cancer classification. Combining FFXO and Bi-GAN enhances prediction accuracy and overcomes key challenges such as overfitting and feature redundancy, leading to a more reliable diagnostic model.

## Data preprocessing

### Missing value imputation

A complex technique for handling missing data sets is MICE (*Van Buuren & Groothuis-Oudshoorn, 2011*). With the exception of qualitative characteristics, unavailable values are imputed for every characteristic in this study using the MICE algorithm. Other attribute categories are labelled as inputs in this technique, whereas the single attribute column with missing data is regarded as an output. Subsequently, the resultant information is predicted iteratively using the regression approach. Every value that is unavailable in the dataset is substituted using a different value in each iteration, and the procedure is repeated until confluence is achieved.

### Feature scaling

The two key methods used in feature scaling are normalization and standardization. The average value is deducted from the pattern of distribution changes and divided by the standard deviation in the standardization process. Scaling is splitting every information point by the standard deviation by taking the average value and eliminating it from all the information points. As outliers are maintained through standardization, the resulting algorithm is less vulnerable to outside effects than when it is not standardized. The mathematical formulation of this process is as shown in Eq. (1),

$$s' = \frac{s - \mu}{\sigma}. \tag{1}$$

In the above equation, $s$ is the original data, and $s'$ denotes the standardized form of the original data, with $\mu$ and $\sigma$ representing the average and the standard deviation values, respectively. The values of $\mu$ and $\sigma$ can be computed using the representations in Eqs. (2) and (3).

$$\mu = \frac{\sum_{k=1}^{T} s}{T} \tag{2}$$

$$\sigma = \sqrt{\frac{1}{T} \sum_{k=1}^{T} (s_k - \mu)^2}. \tag{3}$$

Further, the standard scaler method converts the values in different dimensions to a unique format. After performing standardization, feature values are normalized using the min-max scaler technique. This scaler method restricts the values within specific boundary limits from least to highest. Min-max scaler can be represented mathematically as shown in Eq. (4),

$$s' = \frac{s - s_{min}}{s_{max} - s_{min}}. \tag{4}$$

As per the above equation, the feature values are restricted between zero and one as minimum ($s_{min}$) and maximum ($s_{max}$) values.

*Data splitting*

After normalizing the data set, the final pre-processing stage entailed splitting the available data into two groups: training and testing data. Eighty percent of the accessible information was broken up into training, and twenty percent was set aside for testing. This splitting is essential to train and validate the model and assess the performance of the Bi-GAN classifier. The forecasting of lung disease by the classification algorithm is implemented after conducting an exploratory data analysis (EDA) on the dataset.

## Feature selection using FFXO

The FFXO algorithm is a metaheuristic optimization technique inspired by the natural environment that simulates the feeding habits of big fruit bat species known as flying foxes. These bats have organized exploration strategies; they use group movement and interaction to find food reserves across large distances. This optimization algorithm simulates a collection of flying foxes investigating a solution set to replicate their behavior. It is a potential feature selection technique for lung cancer detection as it can effectively explore high-dimensional pattern regions, enhance classification quality, and manage chaotic data to improve evaluation accuracy. The algorithm begins by generating arbitrary solutions, which are considered to be the location of the flying foxes as represented in Eq. (5),

$$s = (s_1, \ldots, s_k). \tag{5}$$

The flying foxes move towards a cooler location to mitigate the hot temperature. This migration behavior can be formulated as shown in Eq. (6),

$$s_{a,b}^{k+1} = s_{a,b}^k + v.rand(cool_b - s_{a,b}^k). \tag{6}$$

In the above equation, $v$ is an arbitrary value that is fixed in nature, and $cool_b$ denotes the location of the flying fox, which is considered the most optimal solution. Upon finding the coolest location, it also explores the next best location to prohibit strangling. The updated locations are represented as given in Eqs. (7) and (8),

$$curr_{a,b}^{k+1} = s_{a,b}^k + rand_{1,b}.\left(cool_b - s_{a,b}^k\right) + rand_{2,b}.(s_{R_1,b}^k - s_{R_2,b}^k) \tag{7}$$

$$s_{a,b}^{k+1} = \begin{cases} curr_{a,b}^{k+1}, & if\ b = c\ or\ rand_j \geq p \\ s_{a,b}^k, & otherwise \end{cases}. \tag{8}$$

In the Eq. (8), $rand_j$ represents an arbitrary value which lies between zero and one, $p$ denotes a constant value that corresponds to the likelihood factor, $s_{R_1,b}^k$ and $s_{R_2,b}^k$ are members from the group that are generated arbitrarily. $curr_{a,b}^{k+1}$ represents the solutions that are currently generated is ensured that the current solutions are not duplicates of the previously generated solutions. If $curr_{a,b}^{k+1}$ is found to be the optimal solution, then it is set to be the next coolest location. Otherwise, the flying fox is assumed to be in the previous location. The flying foxes die for two reasons, and in such cases, the locations of new flying foxes need to be determined. Firstly, the flying foxes die when the temperature is too high,

so they cannot return to their original locations. The dead flying foxes are replaced by new ones and the location of the replaced flying foxes is determined as given in Eq. (9),

$$s_{a,b}^{k+1} = \frac{\sum_{c=1}^{N} rp_{c,b}^{k}}{n}. \tag{9}$$

Secondly, the flying foxes will also die when circumscribed by other flying foxes in the group. Therefore, the probability of flying foxes ($p_{best}$) to be present in the coolest place is computed using the Eq. (10) as,

$$p_{best} = \frac{num - 1}{tot_{size}}. \tag{10}$$

In the above equation, $num$ denotes the total number of flying foxes that have the same set of optimal solutions and $tot_{size}$ denotes the total population of flying foxes in the search space. The process of copulation between two flying foxes is performed by arbitrarily selecting two members ($G_1, G_2$) as parents to produce new successors. The resultant successors are determined using the representations in Eqs. (11) and (12),

$$successor_1 = H.G_1 + (1 - H).G_2 \tag{11}$$

$$successor_2 = H.G_2 + (1 - H).G_1. \tag{12}$$

$H$ in the above equation, it denotes an arbitrarily generated value that lies within the range of zero and one. The algorithmic steps involved in the FFXO algorithm are presented in Algorithm 1.

Several algorithm parameters in the Bi-GAN + FFXO model affect classification accuracy, training stability, and computational resource efficiency. The learning initial rate (0.0001 with Adam optimizer) provides stabilization by allowing smooth convergence. A batch size of 32 is a good compromise between computation and generalization, preventing overfitting while allowing sufficient stable gradient updates. The number of training epochs is set to 100 to provide reasonable learning, with an early stopping allowance to assist in conserving computation if validation performance has not improved for some time. Feature selection is done using FFXO, where 15 features with most relevance to the classification task are selected. This strategy provides enough dimensionality reduction to maintain classification accuracy. The population size in FFXO (50) limits the number of considered feature subsets and it provides a compromise between computational efficiency and diversity of the search space. In addition, mutation (0.3) and crossover (0.7) rates provide tradeoff selection ensuring diversity in the feature sets selected and a balance between exploration and exploitation. In Bi-GAN with 128 latent vector size, model robustness and data synthetic generation is enhanced, while three hidden layers in the generator and the discriminator networks improve data feature extraction. All these parameters enhance the generalisation capability of the model, along with FFXO, which improves feature selection, and Bi-GAN, which increases feature learning and dataset augmentation. Metaheuristic optimization paired with adversarial training achieves maximised classification accuracy (98.7%) for computational power.

---

**Algorithm 1 Flying fox optimization algorithm.**

**Input:** $N \rightarrow$ Number of flying foxes (population size)

$T \rightarrow$ Maximum number of iterations

*FeatureSet* $\rightarrow$ Initial feature space

*FitnessFunction* $\rightarrow$ Function to evaluate feature subsets

**Output:** *BestFeatureSubset* $\rightarrow$ Optimal selected features

1 **Procedure:**

2 Initialize population of $N$ flying foxes with random feature subsets;

3 Evaluate fitness of each flying fox using *FitnessFunction()*;

4 Determine the best solution (*CoolestLocation*) and worst solution (*HottestLocation*);

5 **for** $t = 1$ to $T$ **do**

6     **for** *each flying fox $F_i$* **do**

7         Move towards *CoolestLocation* using adaptive position update;

8         **if** *$F_i$ finds a better solution* **then**

9             Update its position.

10     **for** *each flying fox reaching extreme conditions (too hot/cold)* **do**

11         Replace it with a new flying fox at a random position;

12     Update *CoolestLocation* and *HottestLocation* based on new fitness evaluations.

13     Perform reproduction (mutation and crossover) among best foxes.

14 Return the *BestFeatureSubset*;

---

Future research may consider hyperparameter tuning strategies like Bayesian optimization or grid search to further optimise these values.

## Classification using Bi-GAN

In a GAN approach, two neural networks—the discriminator and the generator, compete against one another in a game-like fashion, usually taking the structure of a game with no losers in which the success of one entity equals the setback of another entity. The discriminative network, also known as the discriminator, assesses the newly created data points that the generative network pulls from a minimal dimensional range. The discriminator separates the newly generated datasets produced by the generator from the original data distribution, while the generator trains how to transform from random variability to a data distribution of the actual information.

Bi-GANs are primarily known for their ability to encode and decode data, learning rich latent representations of input features. While Bi-GANs are often applied in image generation, data augmentation, and anomaly detection, their unique ability to map data to a latent space makes them highly effective for classification tasks, particularly in medical diagnosis. The primary motivation for using Bi-GAN in this study is its ability to enhance feature representation learning. Traditional deep learning models, such as CNNs and RNNs, rely on manually extracted or predefined features, which can limit their effectiveness in handling complex medical datasets. In contrast, Bi-GAN automatically

learns intricate patterns in the data by mapping it to a structured latent space, which enables the model to differentiate between cancerous and non-cancerous lung tissue more effectively. The encoder network in Bi-GAN captures discriminative features, improving classification performance. Another key reason for using Bi-GAN is its ability to handle class imbalance and improve generalisation. Medical datasets, particularly those related to lung cancer, often suffer from imbalanced class distributions, where there are significantly fewer malignant cases compared to healthy samples. A traditional classifier may become biased toward the dominant class, leading to poor recall and high false negatives, which can be detrimental in a medical context. Bi-GAN enhances the classifier's generalisation ability by generating synthetic data representations, ensuring better discrimination between positive and negative cases.

An encoder is added to the basic GAN model to create an enhanced version known as bidirectional GAN, or Bi-GAN. To better assist the generator in creating artificial datasets that are more conceptually affluent, the Bi-GAN can now learn a reverse translation from the actual information to the hidden space through the incorporation of an encoder. Here, the encoder helps the Bi-GAN model understand the hidden representation of the exact information, an important function of the encoder. Similar to a regular GAN, there are two phases in the training process. The discriminator must first be trained to maximize the goal parameter without changing the generator, as shown in Eq. (13). The generator and encoder are trained to minimise the same desired function connected to the discriminator in the subsequent step. The structure of Bi-GAN is depicted in Fig. 2.

$$
\underbrace{min}_{G,E} \underbrace{max}_{D} H(D, E, G) = E_{a \sim dA} \left[ E_{c \sim dE(\cdot|a)} [log \, D(a, c)] \right]
$$
$$
+ E_{c \sim dC} [E_{a \sim dG(\cdot|c)} [log(1 - D(a, c))]]. \tag{13}
$$

The discriminator determines a single objective function, even though the generator and the discriminator are trained concurrently. Through the discriminator, both the encoder and the generator get passive modifications. In contrast to the conventional GAN method, the discriminator consists of an additive step. The fusion process combines data with its hidden space E(a). The discriminator then receives both of these combined sets of information. During the optimisation, it can be seen that G and E are reversed representations of each other, which are represented as a = G (E(a)) and c = E (G(c)). The algorithm to train the Bi-GAN technique is presented in Algorithm 2.

After identifying and selecting the essential aspects, classification was performed using Bi-GANs. Classifiers like decision trees and support vector machines have difficulty using medical data due to its high dimensionality, class imbalance, and many complex interactions among the features. Such class imbalance problems are often mitigated in the models using Bi-GANs' ability to create synthetic features. Moreover, Bi-GANs enhance the model's robustness. Standard GANs typically consist of a Generator (G) that generates synthetic samples and a Discriminator (D) that attempts to distinguish real samples from the fake ones. Bi-GAN incorporates an Encoder (E) that learns to map real data into a latent vector.

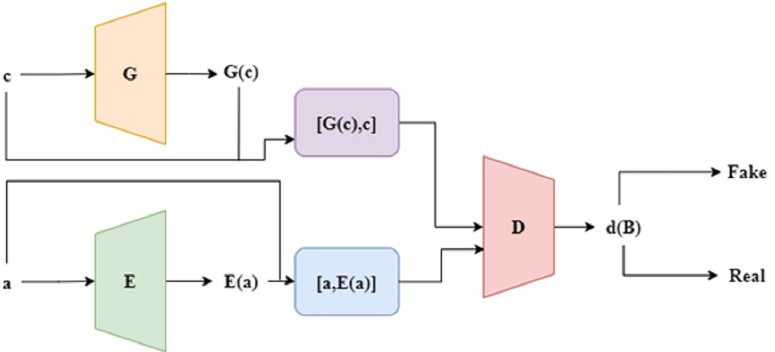

**Figure 2 Structure of Bi-GAN.**

---

**Algorithm 2 Bidirectional generative adversarial networks (Bi-GAN).**

**Input:** *Dataset* → Preprocessed dataset with selected features

*Epochs* → Number of training iterations

*LearningRate* → Learning rate for optimization

*G* (Generator), *E* (Encoder), *D* (Discriminator) → neural networks

**Output:** Trained Bi-GAN Model

1  **Procedure:**

2  Initialize Generator ($G$), Encoder ($E$), and Discriminator ($D$) with random weights;

3  **for** *epoch* = 1 *to Epochs* **do**

4       **Train Discriminator ($D$);**

5       Sample real data $X_{real}$ from the dataset;

6       Generate fake data: $X_{fake} = G(Z)$, where $Z$ is random noise;

7       Encode real data to latent space: $Z_{real} = E(X_{real})$;

8       Compute discriminator loss and update $D$;

9       **Train Generator ($G$) and Encoder ($E$);**

10      Generate synthetic samples: $X_{fake} = G(Z)$;

11      Encode real samples: $Z_{real} = E(X_{real})$;

12      Compute generator loss and update $G$ and $E$;

13  Evaluate the trained Bi-GAN model on the test dataset;

14  Return the final trained model for lung cancer classification;

---

This is done to enhance feature extraction and classification. There are two main steps in the training of Bi-GAN. Initially, the D undergoes training to differentiate between genuine patient records and those produced through the G processes. The Generator learns how to make realistic features of lung cancer, which increases the quality of samples it tries to produce, which closely matches actual patient data. Furthermore, the Encoder (E) model translates actual samples into latent space, which helps in modeling the fundamental structure of lung cancer-associated patterns. This iterative training process

continues in a two-way direction for the generator and the discriminator, ensuring that the generator improves the quality of the data generated. In contrast, the discriminator improves in knowing what information is accurate and what is counterfeit. At the end of the training, the classification of lung cancer using the Bi-GAN model is highly accurate as there is an integrated selection of feature processes *via* FFXO and data generation processes through adversarial learning.

Multiple assumptions were made during the experiment to maintain the accuracy and replicability of the methodology. The model was assumed to perform reasonably well on previously unseen data because the dataset was a representative sample of real-world lung cancer patients. In addition, it was believed that the missing values were missing at random, which justified the use of MICE for imputation. The selected features were chosen using a combination of statistical methods (*i.e.*, correlation matrix) and clinical expertise, ensuring that those chosen features were significant risk factors for lung cancer. In addition, it was assumed that the Bi-GAN model could generalise well to new test data, mainly because the model was put through a rigorous 5-fold cross-validation process.

Although the approach uses deep learning in the scope of lung cancer detection, it incorporates several novel contributions that separate it from other methods. The combination of FFXO with Bi-GAN represents an essential feature selection and classification development. Instead of traditional implementations of deep learning where structure changes or hyperparameter tuning are the only options, this research alters feature selection at the level of metaheuristic optimization, resulting in higher accuracy, lower computational costs, and enhanced generalisation.

Perhaps the most important innovation in this work is using FFXO as a metaheuristic feature selector. Unlike conventional techniques like principal component analysis (PCA) or recursive feature elimination (RFE), FFXO has the capability to selectively and adaptively retrieve relevant attributes by making use of both exploration and exploitation. These types of techniques enable the model to effectively remove irrelevant features, enhance learning, and mitigate overfitting. While feature selection is one of the key components of deep learning models, it tends to be neglected. By using FFXO in this study, the efficiency of classification processes is significantly augmented while still retaining high levels of prediction accuracy.

This study seeks to expand the use of Bi-GAN for detection of lung cancer. This area appears to be pioneering and Bi-GAN approach differs from the classic use of CNNs, SVMs, or decision trees in that it performs classification and also generates synthetic data to address class imbalance problems. Bi-GAN's adversarial framework allows it to capture data patterns and extract features that are more complex, resulting in improved performance of the classification model. The introduction of an encoder in Bi-GAN enables the model to learn deep latent feature representations. This improvement enhances the model's ability to accurately classify lung cancer *vs.* non-cancer cases. This characteristic of the proposed approach is advantageous in situations where there are few labeled medical cases, making it more practical in allowing the model to generalize to novel cases.

## RESULTS AND DISCUSSION

This section presents the results obtained from the experimental evaluation of the proposed system on the lung cancer dataset and compares the outcomes against the conventional approaches in the literature.

### Experimental setup

The experiments were conducted on a Windows-based machine with an Intel® Core™ i5-7200U CPU (2.50 GHz) and 4 GB RAM. The implementation was performed in Python, using deep learning frameworks such as TensorFlow, Keras, and PyTorch, along with essential libraries including Matplotlib, Scikit-learn, and Pandas for data handling and visualization. The dataset used for evaluation was obtained from Kaggle's publicly available lung cancer dataset, which consists of 284 instances and 16 attributes, comprising both categorical and numerical variables. The methodology ensures that proper feature engineering, optimization techniques, and deep learning architectures are employed to maximize classification performance.

### Dataset description

The lung cancer dataset used in this research is composed of 16 attributes, which is a combination of numerical and categorical variables (https://www.kaggle.com/datasets/akashnath29/lung-cancer-dataset). There are a total of 284 instances in the dataset, which aids in making a classification based on the cancer risk status. The dataset employed in the experimentation can be downloaded from the link given. A detailed description of the variables available in the dataset is presented in Table 2.

### Experimental evaluation

One technique that can be used to comprehend the fundamental connections between different data characteristics in a collection of data is feature correlation. Determining the relationships that exist between the data attributes and the manner in which every attribute affects the resulting feature are just two applications where feature correlation can be helpful. In this work, the correlation between the features is computed using the correlation coefficients of a attribute vector $W$ which is of the dimension $m \times n$ represented as $W = [w_1, w_2, \ldots, w_n]$ where $w_1, w_2, \ldots, w_n$ denotes the set of attributes, m denotes the length of each of the attribute and n denotes the total number of attributes. Figure 3 represents the resultant correlation matrix obtained for the given dataset. Based on the correlation analysis, it can be observed that there is a moderate positive correlation (0.29) between alcohol consumption and lung cancer which indicates that an individual with the habit of consuming alcohol is likelier to have lung cancer.

The correlation matrix as shown in Fig. 3 visually represents the relationships between different attributes in the dataset. The analysis reveals that alcohol consumption (0.29 correlation), wheezing and coughing (0.25 correlation), and fatigue (strong correlation) are among the most influential features in lung cancer classification. On the other hand, attributes such as age (0.11 correlation) and gender (0.054 correlation) exhibit weak relationships with lung cancer occurrence. These findings suggest that lifestyle factors and

**Table 2 Detailed description of the variables available in the dataset.**

| S. No | Feature name | Feature description | Type | Possible values |
|---|---|---|---|---|
| 1 | Gender | Denotes the gender of the patient | Categorical | Male/Female |
| 2 | Age | Indicates the age of the patient | Numerical | 1–100 |
| 3 | Smoking | Denotes the smoking habits of the patient | Categorical | Yes/No |
| 4 | Yellow fingers | Represents the presence of yellow fingers | Categorical | Yes/No |
| 5 | Anxiety | Indicates the anxiety level of the patient | Categorical | Yes/No |
| 6 | Peer_pressure | Represents the pressure due to lifestyle changes | Categorical | Yes/No |
| 7 | Chronic disease | Indicates if the patient is affected by any chronic disease | Categorical | Yes/No |
| 8 | Fatigue | Represents the fatigue level of the patient | Categorical | Yes/No |
| 9 | Allergy | Denotes if the patient is inflicted with any form of allergy | Categorical | Yes/No |
| 10 | Wheezing | Denotes the presence of wheezing in the patient | Categorical | Yes/No |
| 11 | Alcohol | Indicates if the patient is an alcoholic | Categorical | Yes/No |
| 12 | Coughing | Represents whether the patient experiences coughing symptoms | Categorical | Yes/No |
| 13 | Shortness of breath | Denotes whether symptoms of shortness of breath are present | Categorical | Yes/No |
| 14 | Swallowing difficulty | Indicates if the patient has issues with swallowing food | Categorical | Yes/No |
| 15 | Chest pain | Represents if the patient is affected with chest pain | Categorical | Yes/No |
| 16 | Lung cancer | Indicates whether the patient is affected with lung cancer or not | Categorical | Positive/Negative |
| 17 | Shortness of breath | Denotes whether the occurrences of symptoms related to shortness of breath | Categorical | Positive/Negative |
| 18 | Wheezing | Represents if the patient if affected with wheezing issues | Categorical | Positive/Negative |
| 19 | Swallowing difficulty | Indicates if the patient has issues with swallowing food | Categorical | Positive/Negative |
| 20 | Clubbing of fingernails | Denotes whether the patient experiences issues related to clubbing of fingernails | Categorical | Positive/Negative |
| 21 | Frequent cold | Represents if the patient if affected with frequent colds | Categorical | Positive/Negative |
| 22 | Dry cough | Indicates if the patient has dry cough problems | Categorical | Positive/Negative |
| 23 | Snoring | Denotes whether the patient has snoring issues | Categorical | Positive/Negative |

respiratory symptoms are stronger indicators of lung cancer risk compared to demographic characteristics. The correlation analysis helps in understanding the underlying patterns in the data, further reinforcing the significance of the FFXO-based feature selection approach in this study.

Features such as wheezing and coughing are moderately correlated (0.25) indicating that these kind of respiratory issues increases the risk of lung cancer. Another essential predictor identified from the correlation matrix is fatigue, which shows a strong correlation with lung cancer. The features having low correlation with lung cancer include age (0.11), gender (0.054) and chronic disease (0.14). Further, strong inter-feature correlations are found between anxiety and fatigue, which is 0.56 and anxiety and wheezing, which is 0.48. Features including fatigue, alcohol consumption, wheezing, and coughing that have stronger associations with lung cancer are probably crucial for lung cancer detection models. However, in this particular dataset, characteristics such as age and gender exhibit weak correlations and may not have as much of an influence on lung cancer prediction. The features selected by the FFXO algorithm (Table 3) include smoking, anxiety, fatigue, allergy, wheezing, coughing, alcohol consumption, shortness of breath, and chest pain, as these attributes exhibited strong correlations with lung cancer risk. The

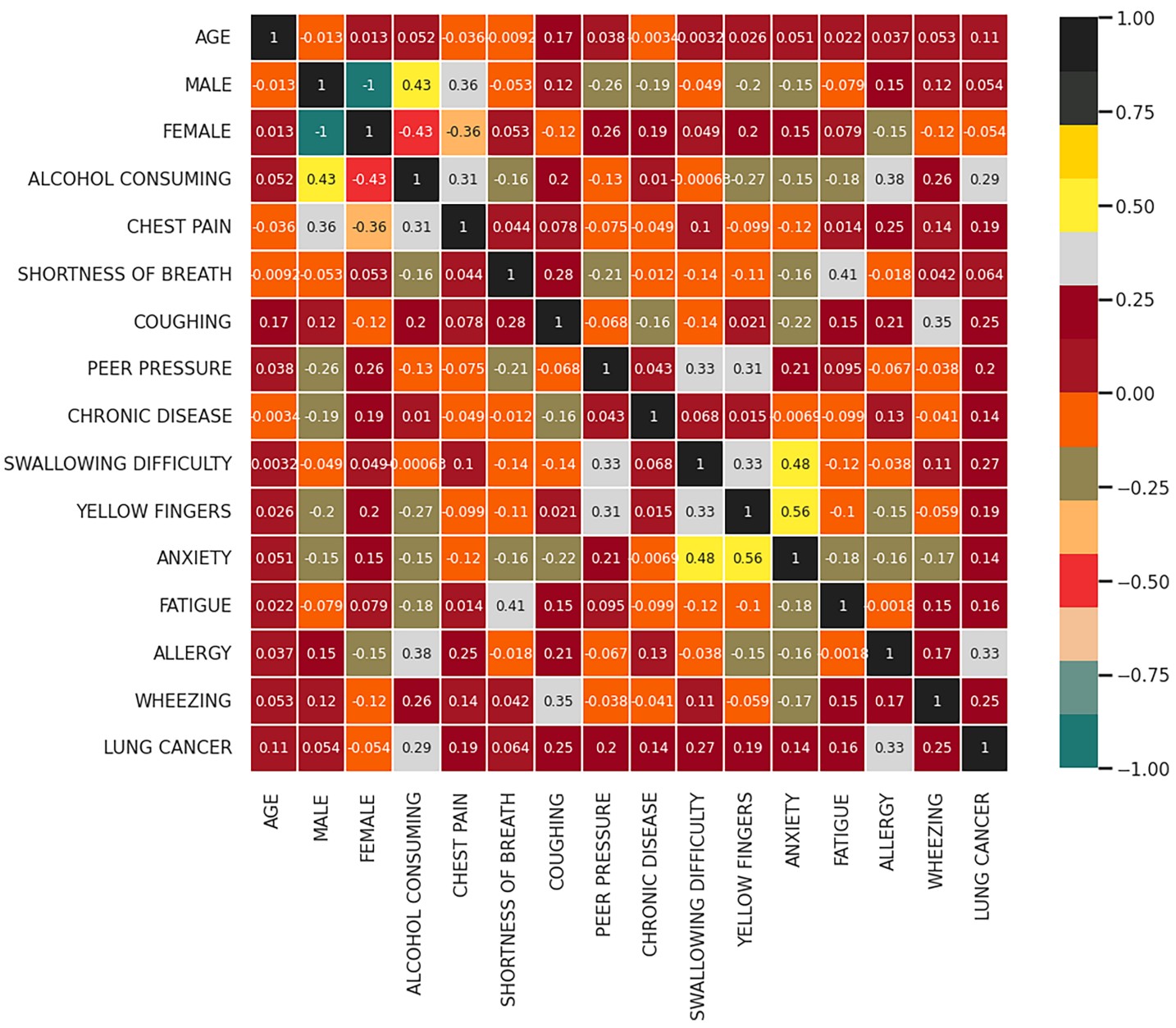

**Figure 3 Correlation matrix.**

selection of these features ensures that the model focuses on the most relevant factors, thereby enhancing classification accuracy while eliminating irrelevant attributes that may introduce noise into the learning process. This feature selection process significantly contributes to the robustness of the proposed approach, as it ensures that only meaningful variables influence the classification outcomes.

Based on the correlation analysis and application of FFXO algorithm, the features selected for classification purposes are presented in Table 3. The features such as smoking,

| S. No | Feature name | Feature description | Type | Possible values |
|-------|--------------|---------------------|------|-----------------|
| | **Table 3 Features selected by FFXO algorithm.** | | | |
| 1 | Smoking | Denotes the smoking habits of the patient | Categorical | Yes/No |
| 2 | Anxiety | Indicates the anxiety level of the patient | Categorical | Yes/No |
| 3 | Fatigue | Represents the fatigue level of the patient | Categorical | Yes/No |
| 4 | Allergy | Denotes if the patient is inflicted with any form of allergy | Categorical | Yes/No |
| 5 | Wheezing | Denotes the presence of wheezing in the patient | Categorical | Yes/No |
| 6 | Alcohol | Indicates if the patient is an alcoholic | Categorical | Yes/No |
| 7 | Coughing | Represents whether the patient experiences coughing symptoms | Categorical | Yes/No |
| 8 | Shortness of breath | Denotes whether the occurrences of symptoms related to shortness of breath | Categorical | Yes/No |
| 9 | Chest pain | Represents if the patient is affected with chest pain | Categorical | Yes/No |
| 10 | Wheezing | Represents if the patient if affected with wheezing issues | Categorical | Yes/No |
| 11 | Swallowing difficulty | Indicates if the patient has issues with swallowing food | Categorical | Yes/No |
| 12 | Frequent cold | Represents if the patient if affected with frequent colds | Categorical | Yes/No |

anxiety, fatigue, allergy, wheezing, coughing, alcohol consumption, shortness of breath and chest pain are considered to be the significant attributes for effective lung cancer categorization.

## JUSTIFICATION FOR MODEL TYPE USED

The Bi-GAN model outshined the rest in lung cancer classification because it had the best performance in terms of accuracy, precision, recall, and F1 score among other models. It is particularly the case with healthcare where the sets of data have more negatives (healthy) to the positives (cancer) and the measures may not be upheld. This is especially the case where the treatment involves cancer detection. Accuracy becomes limited in that Bi-GAN has the great advantage of false positive minimization, and because of high recall, cancer goes undetected the person, which ensures better reliability.

Among the conventional ML and DL models, with or without feature selection, Bi-GAN achieved reasonable accuracy performance and effectiveness across metrics regardless of time and even outperformed recent works in lung cancer recognition. Also, application of FFXO led to an increase of the model's accuracy up to 98.7% which positioned the model as the best. Its intricacy which may lower its practical application in the industry notwithstanding, Bi-GAN guarantees good trade off of its sensitivity and specificity thus classifying lung cancer efficiently.

### Performance assessment

The performance of the proposed system is assessed based on the metrics such as accuracy, F1-score, precision and recall. In a balanced dataset with about identical amounts of lung cancer and non-lung cancer samples, accuracy offers a clear assessment of how effectively the Bi-GAN model distinguishes between the two classes. The information set may be unbalanced in healthcare applications such as lung cancer detection, with a higher number of negative (healthy) instances than positive (cancer) instances. When this occurs, accuracy on its own may be deceptive since a model that classifies the majority of

specimens as negative may yet have high accuracy even when it is unable to identify real instances of cancer. For this reason, other measures like F1-score, recall, and precision are required. A high precision score in the classification of lung cancer indicates that the Bi-GAN model minimises errors in diagnosis, an important feature that helps prevent inaccurate identification of cancer in healthy persons. By ensuring that the majority of lung cancer cases are accurately identified by the model, recall reduces the possibility of false negatives. Since the F1-score strikes a compromise between recall and precision, it is perfect for assessing the Bi-GAN in situations where false positives and false negatives might have serious repercussions.

The performance of the Bi-GAN model used in this research for lung cancer detection is compared against traditional ML and DL models along with optimization techniques as well as with the existing works in the literature. Initially, the performance of the Bi-GAN model is compared with the ML models without employing feature selection techniques and the outcomes are presented in Table 4 and Fig. 4. Decision tree model produced an accuracy of 87.6% with 86.9% F1 score. Random forest model exhibited an accuracy, precision, recall values of 88.9%, 87.4% and 87.6% respectively. Naïve Bayes classifier offered predictions with 90.8% accuracy and 89.6% precision levels. Logistic regression model was capable enough to produce predictions with 91.5% accuracy, 90.3% precision, 89.6% recall and 91.2% F1 score. AdaBoost classifier produced higher accuracy among other models with 92.7% accuracy and 91.3% precision. It was observed that the Bi-GAN model produced maximum accuracy of 94.3%, precision of 93.5%, recall of 93.7% and F1 score of 94.1%.

The comparison of machine learning models without feature selection (Table 4 and Fig. 4) demonstrates that traditional classifiers such as decision tree (87.6%) and random forest (88.9%) produce relatively lower accuracy levels. In contrast, the proposed Bi-GAN model achieves 94.3% accuracy, indicating its superior capability in recognizing lung cancer patterns. However, machine learning models often struggle with high-dimensional data, leading to suboptimal performance. This challenge is addressed in the subsequent experiment, where feature selection is applied before classification.

Further, the ML models were also assessed after applying selected features for performing classification and the results are presented in Table 5 and Fig. 5. It can be seen that the accuracy and other metric values for all the models have increased after using feature selection. The accuracy of decision tree was 89.6% with 88.2% precision value. Random forest and naïve Bayes classifier showed accuracy improvement as 90.6% and 92.3% respectively. Logistic regression model produced an accuracy, precision, recall and F1 score values of 93.6%, 92.1%, 92.4% and 93.3% respectively. AdaBoost classifier showed better accuracy in making predictions with 94.9%. However, the accuracy of Bi-GAN model also improved to 96.7% with precision and recall of 95.3% and 95.7% respectively.

When feature selection is incorporated (Table 5 and Fig. 5), the performance of all classifiers improves, confirming that selecting the most relevant attributes significantly enhances classification accuracy. Bi-GAN achieves 96.7% accuracy, outperforming other models such as AdaBoost (94.9%) and logistic regression (93.6%). This improvement highlights the effectiveness of FFXO in reducing dimensionality, ensuring that the classifier

**Table 4 Performance comparison of state-of-the-art models on the lung cancer dataset.**

| Techniques | Accuracy (%) | Precision (%) | Recall (%) | F1 score (%) |
|---|---|---|---|---|
| Decision Tree | 87.6 | 86.5 | 85.2 | 86.9 |
| Random Forest | 88.9 | 87.4 | 87.6 | 88.3 |
| Naïve Bayes | 90.8 | 89.6 | 89.3 | 90.4 |
| Logistic Regression | 91.5 | 90.3 | 89.6 | 91.2 |
| AdaBoost | 92.7 | 91.3 | 91.7 | 92.4 |
| Multi-Layer Perceptrons (MLP) | 90.6 | 88.6 | 89.2 | 89.8 |
| Deep Neural Networks (DNN) | 92.7 | 90.3 | 91.7 | 92.3 |
| Gated Recurrent Units (GRU) | 93.6 | 92.3 | 92.7 | 93.1 |
| Bayesian Neural Networks (BNN) | 91.8 | 89.8 | 90.6 | 91.4 |
| Deep Belief Networks (DBN) | 95.5 | 93.7 | 93.9 | 94.3 |
| Bi-GAN | 96.7 | 95.3 | 95.7 | 96.4 |

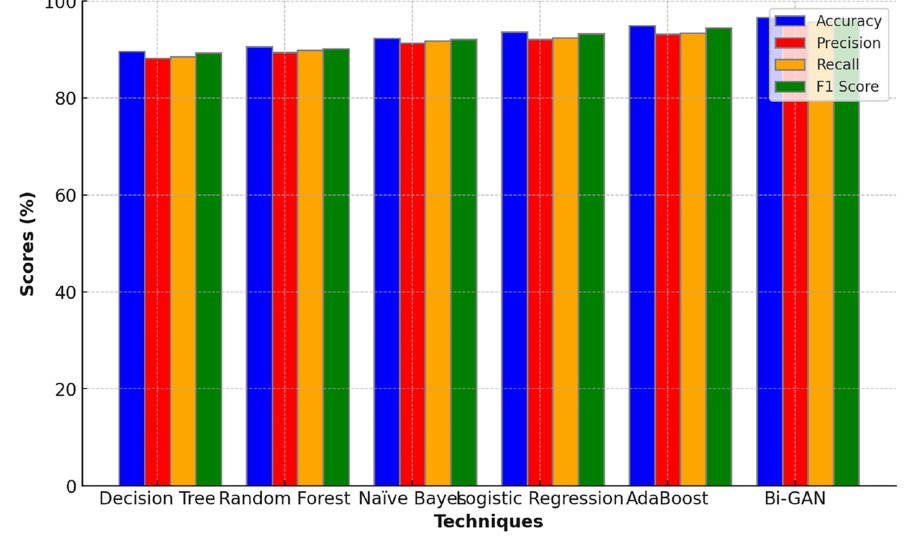

**Figure 4 Comparison with ML models (without feature selection).**

**Table 5 Comparison with ML models (with feature selection).**

| Techniques | Accuracy (%) | Precision (%) | Recall (%) | F1 score (%) |
|---|---|---|---|---|
| Decision Tree | 89.6 | 88.2 | 88.6 | 89.3 |
| Random Forest | 90.6 | 89.4 | 89.8 | 90.2 |
| Naïve Bayes | 92.3 | 91.4 | 91.8 | 92.1 |
| Logistic Regression | 93.6 | 92.1 | 92.4 | 93.3 |
| AdaBoost | 94.9 | 93.2 | 93.4 | 94.5 |
| Bi-GAN | 96.7 | 95.3 | 95.7 | 96.4 |

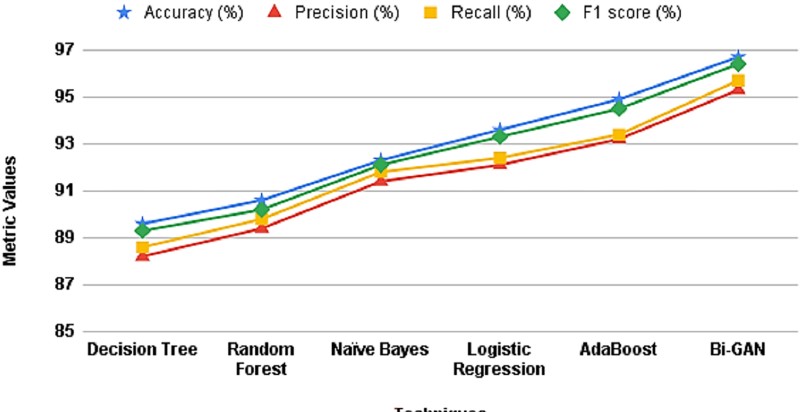

**Figure 5 Comparison with ML models (with feature selection).**

**Table 6 Comparison with baseline models.**

| Techniques | Accuracy (%) | Precision (%) | Recall (%) | F1 score (%) |
|---|---|---|---|---|
| Multi-layer perceptrons | 90.6 | 88.6 | 89.2 | 89.8 |
| Decision Tree (DT) | 87.6 | 86.5 | 85.2 | 86.9 |
| Random Forest (RF) | 88.9 | 87.4 | 87.6 | 88.3 |
| Naïve Bayes (NB) | 90.8 | 89.6 | 89.3 | 90.4 |
| Support Vector Machine (SVM) | 91.2 | 91.0 | 90.5 | 92.3 |
| Deep Neural Networks | 92.7 | 90.3 | 91.7 | 92.3 |
| Gated Recurrent Units | 93.6 | 92.3 | 92.7 | 93.1 |
| Bayesian Neural Networks | 91.8 | 89.8 | 90.6 | 91.4 |
| Deep Belief Networks | 95.5 | 93.7 | 93.9 | 94.3 |
| Bi-GAN | 96.7 | 95.3 | 95.7 | 96.4 |

focuses only on the most important predictors. The results emphasize that a well-optimized feature selection process is crucial in improving lung cancer detection accuracy, making the classification process more efficient.

In addition to this, conventional DL models were also assessed and compared for lung cancer classification. Multi-layer perceptrons (MLP) produced accuracy of 90.6%, precision of 88.6%, recall of 89.2% and F1 score of 89.8%. Deep neural networks (DNN) produced higher accuracy of 92.7% with 90.3% precision value. Gated recurrent units (GRU) produced 93.6%, 92.3% and 92.7% of accuracy, precision and recall values correspondingly. Bayesian neural networks (BNN) showed lower accuracy for lung cancer detection with outcomes accurate with 91.8%. Deep belief networks (DBN) performed better than the other models and produced classifications with 95.5% accuracy. The

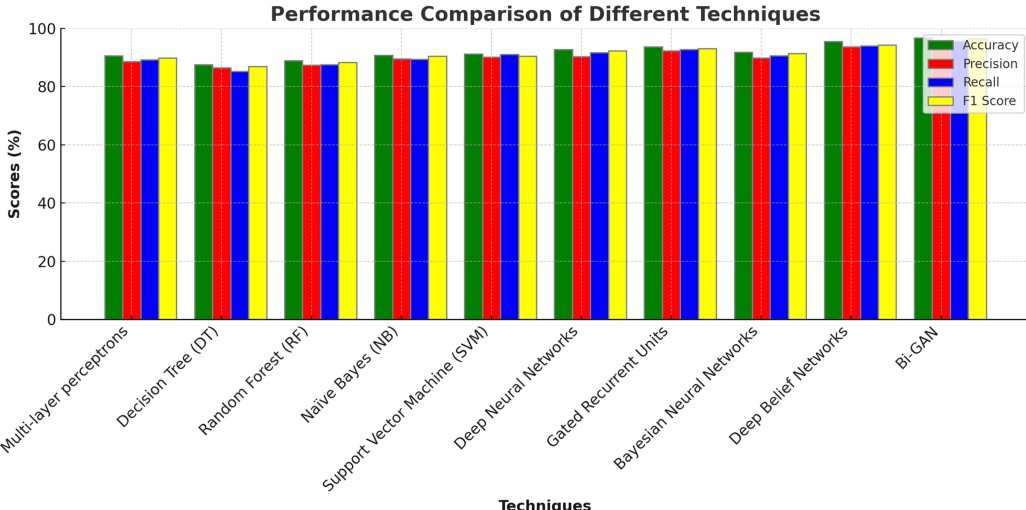

**Figure 6  Comparison with DL models.**

**Table 7  Comparison with optimized DL models.**

| Techniques | Optimization algorithm | Accuracy (%) | Precision (%) | Recall (%) | F1 score (%) |
|---|---|---|---|---|---|
| MLP | GA | 91.9 | 89.8 | 89.3 | 90.5 |
| DNN | PSO | 93.8 | 92.3 | 92.7 | 93.6 |
| GRU | ABC | 94.6 | 93.4 | 93.8 | 94.1 |
| BNN | ACO | 92.3 | 90.6 | 91.4 | 91.8 |
| DBN | CSO | 97.5 | 95.3 | 96.5 | 95.8 |
| Bi-GAN | FFXO | 98.7 | 97.6 | 97.2 | 98.4 |

detailed description of outcomes along with graphical representations are presented in Table 6 and Fig. 6.

The comparison with deep learning models (Table 6 and Fig. 6) further demonstrates the strength of Bi-GAN. Traditional deep learning models such as multi-layer perceptrons (90.6%), deep neural networks (92.7%), and gated recurrent units (93.6%) show competitive performance but still fall short of Bi-GAN's 96.7% accuracy. The superior performance of Bi-GAN is attributed to its ability to generate synthetic data, thereby improving classification robustness and reducing overfitting issues commonly encountered in medical datasets.

The performance of the DL models in combination with different optimization algorithms such as Genetic Algorithm (GA), Particle Swarm Optimization (PSO), Artificial Bee Colony Optimization (ABC), Ant Colony Optimization (ACO) and Cuckoo Search Optimization (CSO) were also analysed. The results of the analysis are shown in Table 7 and Fig. 7. MLP with GA was found to produce an accuracy of 91.9%, precision of 89.8%, recall of 89.3% and F1 score of 90.5%. DNN with PSO algorithm produced improved accuracy of 93.8% with 92.3% precision. GRU with ABC algorithm showed an accuracy enhancement of 94.6% with F1 score value of 94.1%. BNN with ACO exhibited

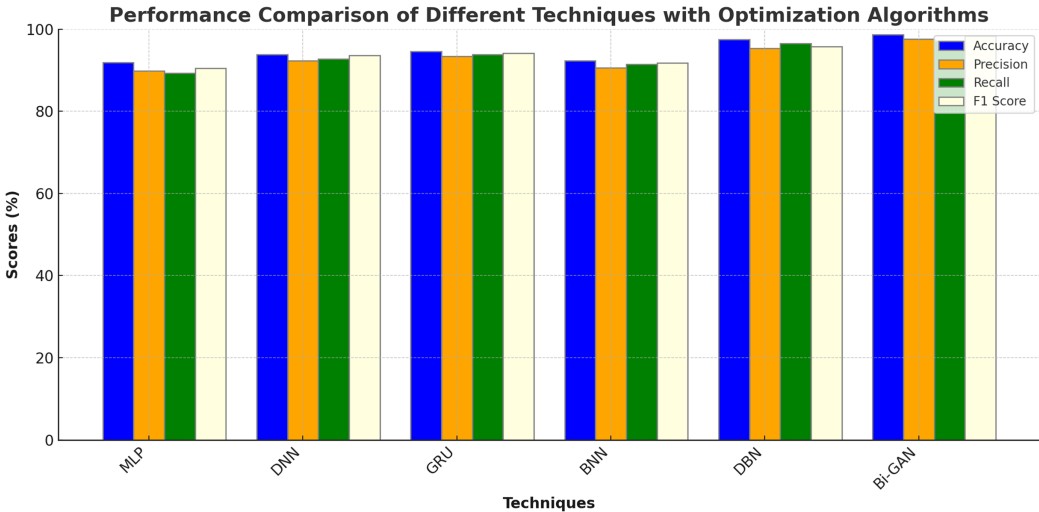

**Figure 7** **Comparison with optimized DL models.**

**Table 8** Comparison with existing works.

| Techniques | Accuracy (%) | Precision (%) | Recall (%) | F1 score (%) |
|---|---|---|---|---|
| *Wang et al. (2022)* | 95.8 | – | – | 94.3 |
| *Mukherjee et al. (2020)* | 94.6 | 93.5 | 92.7 | 94.2 |
| *Mkindu, Wu & Zhao (2023)* | 96.8 | 95.3 | 94.9 | – |
| *Sun, Pang & Li (2023)* | 97.5 | – | – | – |
| *Chen et al. (2021)* | 98.2 | – | – | 97.6 |
| Proposed | 98.7 | 97.6 | 97.2 | 98.4 |

accuracy of 92.3%, precision of 90.6%, recall of 91.4% and F1 score of 91.8%. DBN model showed much higher accuracy of 97.5% when optimized by CSO algorithm. Bi-GAN model with FFXO algorithm produced superior accuracy of 98.7% compared to conventional ML and DL algorithms with and without feature selection methods.

Finally, few research on lung cancer detection using deep learning approaches that are proposed by other researchers are compared with the outcomes of present research and the results are tabulated and depicted in Table 8 and Fig. 8. *Wang et al. (2022)* suggested the dual-modality deep learning network for classifying lung tumors and produced an accuracy of 95.8%. The research in *Mukherjee et al. (2020)* used shallow convolutional neural network to forecast the presence of lung carcinoma and the outcomes were 94.6% accurate with 93.5% precision and 92.7% recall. The research work recommended in *Mkindu, Wu & Zhao (2023)* 3D multi-scale vision transformer was capable enough to detect tumours in lung nodules with 96.8% efficiency. An accuracy level of 97.5% was achieved by applying Improved Swin Transformer for classifying lung cancer in *Sun, Pang & Li (2023)*. Multiple attention 3D U-Net was utilised for segmenting lung tumors in

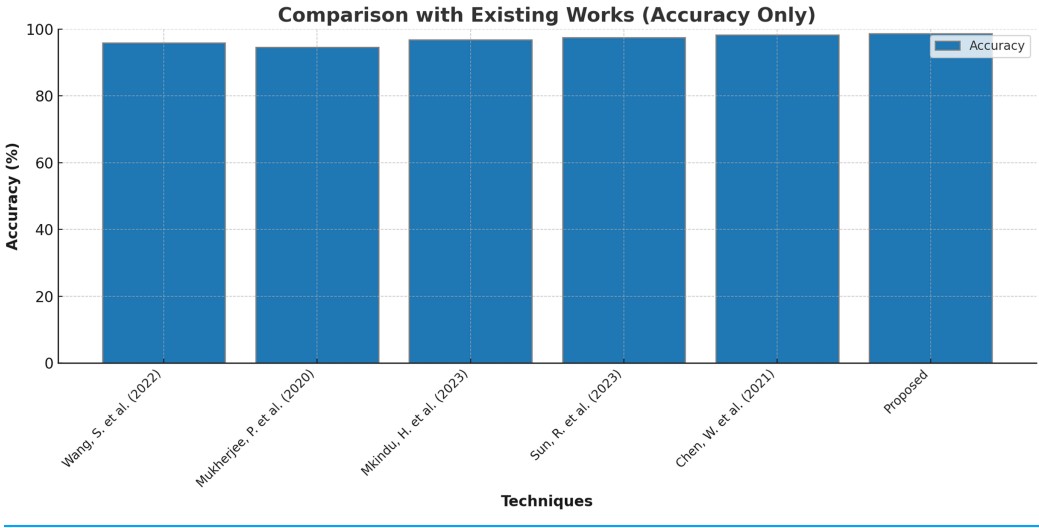

**Figure 8 Comparison with existing works.**

*Chen et al. (2021)* and obtained an accuracy of 98.2% with 97.6% F1 score. The maximum accuracy of 98.7% is produced by the Bi-GAN model used in this research for lung cancer classification.

## Limitations of current research

A highly intricate model may result from combining the FFXO algorithm with deep learning approach like Bi-GAN. High model complexity might make it more challenging to operate in real-time or resource-constrained situations, such as healthcare facilities or environments with limited resources, by increasing computing costs, storage utilisation, and training times. Insufficient or skewed data may restrict the model's usefulness and cause poor generalisation to unobserved cases.

## CONCLUSION

Lung cancer remains a global health concern, where the late detection of the disease assures a high mortality rate. The combination of techniques using deep learning and optimization method has shown some potential in improving early detection, however issues like, high-dimensional data, class imbalance, and inefficiencies in computation optimisation remain. This research incorporates a novel hybrid framework model that combines Bi-GAN for classification with FFXO for features classification to address these problems. The proposed model achieves better classification accuracy, improves feature selection efficiency, and provides better generalisation for smaller datasets. The experimental results proved that the Bi-GAN + FFXO model surpassed the rest by achieving an accuracy of 98.7%. It outperformed the traditional deep learning classifiers, and the modern-day optimisation techniques. The model also effectively solves the class imbalance problem using Bi-GAN for synthetic data generation and quickly classifies small datasets. Furthermore, the FFXO based feature selection approach lowered the dimension

of the dataset enough to lessen the computational load while improving the interpretability of the data compared to the traditional methods like PCA or Autoencoders.

Aside from enhanced accuracy, this study has practical aspects that pertain to clinical implementation. Its computational efficiency enables deployment in resource-limited medical settings where AI-based diagnostic tools must trade accuracy for feasibility. The optimized feature selection guarantees that only a few relevant medical attributes are used to achieve high classification performance, reducing training time and inference latency. Furthermore, the dominance of the proposed methodology was accentuated by the comparison with state-of-the-art deep learning structures and metaheuristic optimisation methods.

### Funding

This work was funded by the Korea Environmental Industry & Technology Institute (KEITI), with a grant funded by the Korean government, Ministry of Environment (The development of IoT-based technology for collecting and managing big data on environmental hazards and health effects), Grant RE202101551. The Deanship of Research and Graduate Studies at King Khalid University funded this work through Large Research Project under grant number RGP2/267/45. Princess Nourah bint Abdulrahman University funded this work through Researchers Supporting Project number PNURSP2025R330. King Saud University funded this work through Researchers Supporting Project number RSPD2025R838. The Deanship of Scientific Research at Northern Border University, Arar, KSA funded this research work through project number NBU-FFR-2025-1564-04. The funders had no role in study design, data collection and analysis, decision to publish, or preparation of the manuscript.

### Grant Disclosures

The following grant information was disclosed by the authors:
Korea Environmental Industry & Technology Institute (KEITI).
Korean government, Ministry of Environment: RE202101551.
The Deanship of Research and Graduate Studies at King Khalid University: RGP2/267/45.
Princess Nourah bint Abdulrahman University: PNURSP2025R330.
King Saud University: RSPD2025R838.
The Deanship of Scientific Research at Northern Border University, Arar, KSA: NBU-FFR-2025-1564-04.

### Competing Interests

The authors declare that they have no competing interests.

### Author Contributions

- Manal Abdullah Alohali conceived and designed the experiments, analyzed the data, prepared figures and/or tables, and approved the final draft.

- Hamed Alqahtani conceived and designed the experiments, analyzed the data, prepared figures and/or tables, authored or reviewed drafts of the article, and approved the final draft.
- Shouki A. Ebad conceived and designed the experiments, analyzed the data, prepared figures and/or tables, and approved the final draft.
- Faiz Abdullah Alotaibi performed the experiments, performed the computation work, prepared figures and/or tables, authored or reviewed drafts of the article, and approved the final draft.
- Venkatachalam K. performed the experiments, performed the computation work, authored or reviewed drafts of the article, and approved the final draft.
- Jaehyuk Cho performed the experiments, performed the computation work, authored or reviewed drafts of the article, and approved the final draft.

## Data Availability

Data is available in the Supplemental Files and at Kaggle: https://www.kaggle.com/datasets/yusufdede/lung-cancer-dataset.

## Supplemental Information

Supplemental information for this article can be found online at http://dx.doi.org/10.7717/peerj-cs.2853#supplemental-information.

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
