# Peer review of "Optimized deep learning approach for lung cancer detection using flying fox optimization and bidirectional generative adversarial networks"

_PeerJ Computer Science, doi:10.7717/peerj-cs.2853_

## Round 0.1 · original submission · Major Revisions

The authors must seriously improve the paper according to reviewers' suggestions.

·

Basic reporting

I have reviewed the article titled "Optimized Deep Learning Approach for Lung Cancer Detection Using Flying Fox Optimization and Bidirectional Generative Adversarial Networks." After carefully examining the manuscript, I have the following suggestions for improvement:

Quality of Figures: Several figures in the manuscript appear blurred, which hinders their clarity and effectiveness. I recommend enhancing their resolution to ensure they are sufficiently clear for readers to interpret the data accurately.

Abstract Enhancement: The abstract should be revised for better clarity and comprehensiveness. It would benefit from a clearer summary of the key objectives, methods, and findings, allowing readers to easily understand the study's contribution.

Algorithm Presentation in Pseudocode: The algorithms presented in the manuscript would be more accessible if written in pseudocode format. This would help readers follow the methodology without needing to decipher programming syntax.

Explanations for Tables and Figures: Some tables and figures could benefit from more detailed explanations. I suggest expanding the descriptions to ensure readers can fully comprehend the results and their significance.

Expansion of Methodology Section: The methodology section is currently somewhat brief. Providing more in-depth details on the experimental setup, methodology rationale, and assumptions would enhance the clarity and reproducibility of the study.

Updating the Literature Review: The literature review should be updated to include the latest and most relevant studies. This would ensure the manuscript reflects current advancements in the field and situates the proposed work within a broader context.

State-of-the-Art Comparison: I recommend including a comparison between the proposed model and existing state-of-the-art techniques, particularly using the same dataset. This would help highlight the strengths and limitations of the proposed method in comparison to current approaches.

I believe addressing these suggestions will improve the overall quality of the manuscript and provide clearer insights for the readers.

Experimental design

Algorithm Presentation in Pseudocode: The algorithms presented in the manuscript would be more accessible if written in pseudocode format. This would help readers follow the methodology without needing to decipher programming syntax.

Validity of the findings

State-of-the-Art Comparison: I recommend including a comparison between the proposed model and existing state-of-the-art techniques, particularly using the same dataset. This would help highlight the strengths and limitations of the proposed method in comparison to current approaches.

Additional comments

Please proof read and avoid grammar mistakes.

Reviewer 2 ·

Basic reporting

The proposed approach only tries to apply DL algorithms on the presented problem.
No insightful methods were proposed

Experimental design

Bidirectional generative adversarial networks are used to "decode" data, why were they chosen for classification?
Authors should justify the specific use of the two feature scaling techniques and MICE for missing value imputation
Why didn't the authors split a prediction part?
The Flying Fox may slow down the feature selection process and not present good performance compared to advanced techniques and models such as auto-encoders
Dataset size is relatively small; thus, the proposed model may not give good performance when it is generalized

Validity of the findings

* The models used for comparison are conventional. Authors should compare their solution with the state-of-the-art solutions

Additional comments

Some figures are not clear

Reviewer 3 ·

Basic reporting

- The given abstract lacks clarity and precision in terms of reflecting the paper's content and findings. It is therefore recommended that the abstract be carefully rewritten to effectively demonstrate the necessity, novelty, and contribution of the research, as well as highlight its major findings.
- The introductory section must appropriately highlight the motivations and objectives of this research. Research gaps and contributions should be better linked to the literature review.
- In the literature review, it is suggested to add the latest frontiers of literature and analyze their main contributions as well as their value and usefulness for this study.
- Regrettably, neither a conceptual comparison with existing approaches nor any discussion about the benefits and drawbacks of the new approach has been put forth. Therefore, discussions and comparative analyses ought to be incorporated. It is imperative to compare the proposed method with the literature.
- The conclusion part must be extended. The conclusion part should contain the author's main findings; and contribution as well as it should give a direction for future researchers. The key contributions of the authors are not highlighted in the present manuscript.
- The motivation for the prospects section in the manuscript is not clearly conveyed. It is noteworthy that your approach your method could be extended to various real-life decision-making problems.

- It may be beneficial to include a brief discussion on the selection criteria for the specific machine learning algorithms used, as this could provide additional context for your choices.
- Check the mathematical notation, especially for the proposed method. This will facilitate the new readers' tracking and applying of the proposed method and get the same results.
-Various figures are not explained well. I suggest adding a brief description of each figure in their captions.
- A complete description of the experimental protocol is missing.
- There are many algorithm parameters in the proposed method. What's the influence of these parameters?

Experimental design

A complete description of the experimental protocol is missing.

Validity of the findings

Regrettably, neither a conceptual comparison with existing approaches nor any discussion about the benefits and drawbacks of the new approach has been put forth. Therefore, discussions and comparative analyses ought to be incorporated. It is imperative to compare the proposed method with the literature.

Additional comments

The paper lacks a comparison with non-deep-learning-based techniques, which could provide a broader context for the proposed solution's strengths and weaknesses."

---

## Round 0.2 · accepted · Accept

The paper was well improved, do it can be accepted.

Reviewer 3 ·

Basic reporting

- The authors have adequately answered all the issues raised, and in my opinion the paper has greatly improved in its fairness and clarity. At this point in my opinion it is ready for publication

Experimental design

- The authors have adequately answered all the issues raised.

Validity of the findings

- The authors have adequately answered all the issues raised.

Additional comments

- The authors have adequately answered all the issues raised.